# External validity of prevalence estimates from the national maternity surveys in England: The impact of response rate

**Sian Harrison**[ORCID]*, **Fiona Alderdice, Maria A. Quigley**

NIHR Policy Research Unit in Maternal and Neonatal Health and Care, National Perinatal Epidemiology Unit, Nuffield Department of Population Health, University of Oxford, Old Road Campus, Headington, Oxford, United Kingdom

* sian.harrison@npeu.ox.ac.uk

## Abstract

### Background

Prevalence estimates from surveys with low response rates are prone to non-response bias if respondents and non-respondents differ on the outcome of interest. This study assessed the external validity of prevalence estimates of selected maternity indicators from four national maternity surveys in England which had similar survey methodology but different response rates.

### Methods

A secondary analysis was conducted using data from the national maternity surveys in 2006 (response rate = 63%), 2010 (response rate = 54%), 2014 (response rate = 47%) and 2018 (response rate = 29%). Unweighted and (for the 2014 and 2018 surveys) weighted survey prevalence estimates (with 95%CIs) of caesarean section, preterm birth, low birthweight and breastfeeding initiation were validated against population-based estimates from routine data.

### Results

The external validity of the survey estimates varied across surveys and by indicator. For caesarean section, the 95%CIs for the unweighted survey estimates included the population-based estimates for all surveys. For preterm birth and low birthweight, the 95%CIs for the unweighted survey estimates did not include the population-based estimates for the 2006 and 2010 surveys (or the 2014 survey for preterm birth). For breastfeeding initiation, the 95%CIs for the unweighted survey estimates did not include the population-based estimates for any survey. For all indicators, the effect of weighting (on the 2014 and 2018 survey estimates) was mostly a shift towards the population-based estimates, yet the 95%CIs for the weighted survey estimates of breastfeeding initiation did not include the population-based estimates.

**Data Availability Statement:** The data underlying this study is not publicly available because the scope of the consent obtained from study participants restricts our ability to share the data on

ethical and legal grounds. There are also third party restrictions by the Office for National Statistics (ONS). Requests to access birth registration data can be submitted to the ONS at https://www.ons. gov.uk/aboutus/whatwedo/statistics/ requestingstatistics/makingarequest; information about the ONS data sharing policy can be found at https://cy.ons.gov.uk/aboutus/ transparencyandgovernance/datastrategy/ datapolicies/onsresearchanddataaccesspolicy. Requests to carry out further analyses on the data from the national maternity surveys can be submitted to the Director of the NPEU at general@npeu.ox.ac.uk. All requests would be subject to the National Perinatal Epidemiology Unit Data Access Policy and may require further regulatory approvals.

**Funding:** This research is funded by the National Institute for Health Research (NIHR) Policy Research Programme, conducted through the Policy Research Unit in Maternal and Neonatal Health and Care, PR-PRU-1217-21202. The views expressed are those of the author(s) and not necessarily those of the NIHR or the Department of Health and Social Care.

**Competing interests:** The authors have declared that no competing interests exist.

## Conclusion

There were no clear differences in the external validity of prevalence estimates according to survey response rate suggesting that prevalence estimates may still be valid even when survey response rates are low. The survey estimates tended to become closer to the population-based estimates when weights were applied, yet the effect was insufficient for breastfeeding initiation estimates.

## Introduction

National maternity surveys have been conducted in England by the National Perinatal Epidemiology Unit (NPEU) since 2006. The national maternity surveys are large cross-sectional postal surveys which explore the health and care of mothers and babies during pregnancy, labour and birth, and the postpartum period. The surveys use the Office for National Statistics (ONS) to obtain samples representative of the target population on key demographic characteristics. Findings from the surveys have contributed to the planning, design and safe delivery of maternity services throughout England [1].

Response rates to the national maternity surveys have dropped from over sixty per cent in 2006 to below thirty per cent in 2018 [2]. This declining trend has been evident in other surveys into maternal and infant health [2–4], and also in surveys on other populations [5]. Different methodological strategies for increasing response rates have been proposed, for example, incentivising participants, generating reminders and keeping surveys short [6] yet, even when these strategies have been implemented, response rates to surveys remain low [7].

Survey response rates are used as an indicator of survey quality and findings from surveys with low response rates can be subject to criticism regarding validity [8]. Another important quality marker is non-response bias, which occurs when those who respond to a survey differ from those who do not on the outcome of interest [9]. This can be particularly problematic in health surveys due to the potential 'healthy volunteer effect' whereby individuals who are more health conscious are more likely to participate [10]. This was found in the UK Biobank whereby participants were less likely to be obese, to smoke, and to drink alcohol on a daily basis and had fewer self-reported health conditions [10]. It is important to understand the magnitude of selective non-response in population-based surveys and the impact this may have on the external validity of prevalence estimates [11]. The existing body of evidence suggests no consistent relationship between response rate and non-response bias in surveys [12–15], yet there have been few large rigorously designed surveys to provide insight into the consequences of lower response rates [16].

The challenge of declining response rates in epidemiological studies requires both the development of innovative recruitment and retention techniques and also the application of statistical methods for adjustment of potential bias introduced by non-response [5]. One such statistical method is the application of survey weights, which involves adjusting the sample of respondents to better represent the characteristics of the sampling frame or the target population. Survey weights are usually derived using data on non-respondents (if available) or the wider population from which the sample was drawn (i.e. post-stratification using census data) [17]. The importance of using weights in reporting on population-based surveys to derive the best estimates of unbiased population prevalence has been highlighted in other national surveys [11].

This study aimed to examine the external validity of prevalence estimates of selected maternity indicators from four national maternity surveys which had similar survey methodology

but different rates of response, by comparing unweighted and weighted estimates from each survey (survey estimates) with population-based estimates available from national routine data (population-based estimates).

The study objectives were:

1. To compare the demographic characteristics of respondents and non-respondents in the 2006, 2010, 2014 and 2018 national maternity surveys

2. To identify the effect of demographic characteristics on response rates and to use the characteristics associated with response to derive survey weights in the 2014 and 2018 national maternity surveys

3. To evaluate the external validity of unweighted and weighted estimates of selected maternity indicators from the 2006 (unweighted only), 2010 (unweighted only), 2014 and 2018 national maternity surveys (survey estimates), compared to population-based estimates from routine data (population-based estimates).

## Methods

### Design and sample

A secondary analysis was conducted using data from the national maternity surveys in 2006 ('Recorded Delivery') [18], 2010 ('Delivered with Care') [19], 2014 ('Safely Delivered') [20] and 2018 ('You and Your Baby') [21]. The national maternity surveys are large cross-sectional postal surveys carried out in England at regular intervals by the National Perinatal Epidemiology Unit (NPEU). The surveys explore the health and care of mothers and babies during pregnancy, labour and birth, and the postpartum period. The samples of women are drawn at random from birth registration records held by the Office for National Statistics (ONS) and are representative of the target population: all women giving birth in England during a specified one-to-two-week period.

For the 2006 survey, 4,800 women who gave birth in March 2006 were sampled, and data on response outcome and demographic characteristics were available for 4,727 women. Of these, 2,966 women responded to the survey, a response rate of 63%. For the 2010 survey, 10,000 women who gave birth in October 2009 were sampled, and data on response outcome and demographic characteristics were available for 9,851 women. Of these, 5,332 responded to the survey, a response rate of 54%. For the 2014 survey, 10,002 women who gave birth in January 2014 were sampled, and data on response outcome and demographic characteristics were available for 9,786 women. Of these, 4,568 women responded to the survey, a response rate of 47%. Finally, for the 2018 survey, 16,000 women who gave birth in October 2017 were sampled, and data on response outcome and demographic characteristics were available for 15,528 women. Of these, 4,509 women responded to the survey, a response rate of 29%. In 2006, 2010 and 2014, women were recruited when they were three months postpartum and, in 2018, women were recruited when they were six months postpartum, and this is likely to have contributed to the marked decrease in response rates between 2014 and 2018 [2].

Ethical approval was obtained separately for each national maternity survey. The most recent survey in 2018 was approved by the London Bloomsbury NRES Committee (18/LO/0271). The return of completed questionnaires was taken as indicating consent to participate.

### Statistical methods

Data on the demographic characteristics of all women selected for the 2006, 2010, 2014 and 2018 surveys (respondents and non-respondents) were provided by ONS. Aggregate data only

were available for the 2006 and 2010 surveys whereas individual-level data were available for the 2014 and 2018 surveys. The demographic data included: age group (16–19, 20–24, 25–29, 30–34, 35–39 or 40+ years); marital status at birth registration (married, joint registration by both parents living at the same address, joint registration by both parents living at different addresses, or sole registration); mother's place of birth (individual country codes); area-based deprivation based on index of multiple deprivation (IMD) for mother's address (grouped into quintiles); region of residence (grouped into nine regions); sex of baby (male or female (2006, 2014 and 2018 only)); multiplicity (singleton or multiple birth (2014 and 2018 only)); and parity (primiparous or multiparous (2018 only)).

All analysis was performed separately for each survey. The differences between the demographic characteristics of respondents and non-respondents to each of the surveys were compared using Chi-Square tests. The association between each of the demographic characteristics and response to the 2006, 2010, 2014 and 2018 surveys was explored (and is presented using bar charts). For the 2014 and 2018 surveys (where individual-level data were available), the association between each of the demographic characteristics and response was also explored using logistic regression. The demographic characteristics which were significantly associated with response (p<0.10) at univariable level were entered into a multivariable model. The final model included only those variables that were significantly associated (p<0.05) with response after adjustment for other variables in the model. The resulting coefficients (adjusted log odds ratios) from the model were used to calculate survey weights for the 2014 and 2018 surveys.

Unweighted prevalence estimates of selected maternity indicators with high levels of completion (caesarean section, preterm birth (gestational age less than 37 weeks), low birthweight (less than 2,500 grams), and breastfeeding initiation) were estimated from each survey (unweighted survey estimates). Weighted prevalence estimates of these maternity indicators were also estimated from the 2014 and 2018 surveys (weighted survey estimates). The estimates were proportions with 95% confidence intervals (CI). We (attempted to) obtain estimates of the same indicators in the target population for each survey (i.e. women who gave birth in: March 2006, October 2009, January 2014 and October 2017). Here we used published estimates of the indicators based on national routine data for all births in England in 2006, 2009, 2014 and 2017. These population-based estimates were used to assess the external validity of the unweighted and weighted survey estimates.

The routine data on mode of birth are reported by NHS Digital (Hospital Episode Statistics (HES)) and the routine data on gestational age at birth and birthweight are reported by ONS. These data have high levels of completion and are considered 'gold standard', thus providing reliable external validation of the survey estimates of these maternity indicators. The routine data on breastfeeding initiation are currently reported by NHS Digital (Maternity Services Dataset (MSDS)) and were formerly reported by the Department of Health (DH). The MSDS was new in 2014 and is based on a subset of all registered births. Breastfeeding initiation status is reported by geographical area and those areas where a valid status is known for fewer than 80% of records do not have a statistic published [22]. Thus, NHS Digital advise caution in interpreting data from the MSDS as an estimate for the whole of England. In addition, the breastfeeding initiation indicator is calculated by dividing the number of live babies born in the period whose first feed is known to be breastmilk by the number of live babies born in the period (and then multiplied by 100) [22]. Therefore, the denominator implicitly assumes that all infants whose breastfeeding status is unknown were not breastfeeding, resulting in an underestimate of the true rates. Due to these caveats, the population-based estimates on breastfeeding initiation are not necessarily considered 'gold standard' and the extent to which they provide reliable external validation of the survey estimates of this indicator is uncertain. Therefore, we used the published population-based estimates in our analysis as well as 'adjusted

population-based estimates' (excluding infants whose breastfeeding status was unknown) which were calculated from the routine data on breastfeeding initiation.

## Results

### Characteristics of respondents and non-respondents

Table 1 shows the distribution of demographic characteristics for the respondents and non-respondents to the 2006, 2010, 2014 and 2018 surveys. The distributions of age group, marital status at birth registration, place of birth, IMD, region of residence, and parity (2018 only) differ in the respondents and non-respondents (p<0.001). Across all surveys, the women who responded were more likely to be older, married when they registered the birth of their baby, born in the UK, living in more advantaged areas, living in the South or East of England, and primiparous (2018 only) compared to the women who were invited to take part but who did not respond. Therefore, the respondents to each survey were not representative of the target population on these demographic characteristics. The distributions of sex of baby and multiplicity were similar for respondents and non-respondents in the surveys for which these data were available (p>0.6).

### Effect of demographic characteristics on response rates

The data in Table 1 were used to describe response rates according to demographic characteristics in each of the surveys (Fig 1). It is clear from Fig 1 that the distribution of responses across the different characteristics was similar for all surveys. For example, the response rate increased with each increase in age group and decreased with each decrease in IMD quintile (indicating higher socioeconomic deprivation) in all surveys. It is also apparent from Fig 1 that the response rate was highest for all subgroups of women across all demographic characteristics in the 2006 survey (horizontal striped columns) and lowest for all subgroups of women in the 2018 survey (plain columns).

Table 2 shows the crude and adjusted odds ratios with 95% CIs for the univariable and multivariable association between demographic characteristics and response to the 2014 or 2018 survey. With the exception of sex of baby and multiplicity, all characteristics were associated with response at univariable level (p<0.001) and were entered into the multivariable logistic regression. With the exception of region of residence in the 2014 survey, all demographic characteristics remained significantly associated with response at multivariable level after adjusting for all other characteristics (p<0.001). Therefore, age, marital status at birth registration, country of birth and IMD were used to calculate the survey weights for the 2014 survey; and age, marital status at birth registration, country of birth, IMD, region of residence, and parity were used to calculate the survey weights for the 2018 survey.

Table 3 shows the unweighted and weighted distributions of the demographic characteristics used to calculate the survey weights for the respondents to the 2014 and 2018 surveys. As expected, applying the weights ensured that the respondents to the surveys were representative of all women giving birth in England during the same time-periods in terms of these demographic characteristics.

### External validity of survey estimates of selected maternity indicators

Table 4 and Figs 2–5 show how the survey estimates of the selected maternity indicators from the 2006, 2010, 2014 and 2018 surveys compare with the population-based estimates pertaining to the same time-periods from national routine data (ONS or NHS Digital (formerly DH for breastfeeding initiation)). The proportions of missing data are low for all population-based

**Table 1. Comparison of respondents and non-respondents to the 2006, 2010, 2014 and 2018 national maternity surveys.**

| Year of survey | 2006 | | | 2010 | | | 2014 | | | 2018 | | |
|---|---|---|---|---|---|---|---|---|---|---|---|---|
| | Respondents (N = 2,966) | Non-respondents (N = 1,865) | p value | Respondents (N = 5,332) | Non-respondents (N = 4,526) | p value | Respondents (N = 4,568) | Non-respondents (N = 5,218) | p value | Respondents (N = 4,509) | Non-respondents (N = 11,019) | p value |
| | % | % | | % | % | | % | % | | % | % | |
| **Age group** | | | p<0.001 | | | p<0.001 | | | p<0.001 | | | p<0.001 |
| <25 years | 20.7 | 32.7 | | 17.1 | 33.5 | | 14.0 | 26.8 | | 9.3 | 18.9 | |
| 25–29 years | 24.9 | 27.6 | | 25.8 | 29.2 | | 26.9 | 29.6 | | 23.4 | 28.3 | |
| 30–34 years | 32.0 | 23.0 | | 32.6 | 22.3 | | 34.7 | 26.1 | | 38.0 | 31.7 | |
| 35+ years | 22.4 | 16.7 | | 24.5 | 15.1 | | 24.4 | 17.5 | | 29.3 | 21.1 | |
| **Marital status** | | | p<0.001 | | | p<0.001 | | | p<0.001 | | | p<0.001 |
| Married | 62.5 | 51.1 | | 61.5 | 47.3 | | 60.1 | 45.8 | | 63.5 | 48.5 | |
| Joint registration (same address) | 27.7 | 27.9 | | 29.1 | 31.1 | | 30.5 | 31.6 | | 29.3 | 33.0 | |
| Other | 9.7 | 20.7 | | 9.6 | 21.5 | | 9.4 | 22.6 | | 7.1 | 18.6 | |
| **Country of birth** | | | p<0.001 | | | p<0.001 | | | p<0.001 | | | p<0.001 |
| UK | 83.8 | 69.5 | | 78.4 | 70.3 | | 76.3 | 70.2 | | 77.2 | 69.1 | |
| Non-UK | 16.2 | 30.5 | | 21.6 | 29.7 | | 23.7 | 29.7 | | 22.8 | 30.9 | |
| **IMD** | | | p<0.001 | | | p<0.001 | | | p<0.001 | | | p<0.001 |
| 1st (most deprived) | 20.3 | 37.6 | | 20.5 | 35.9 | | 19.5 | 34.1 | | 15.7 | 30.8 | |
| 2nd | 19.5 | 22.4 | | 19.0 | 24.2 | | 21.4 | 24.1 | | 19.3 | 23.0 | |
| 3rd | 21.2 | 15.9 | | 21.2 | 17.1 | | 20.4 | 17.0 | | 21.0 | 18.0 | |
| 4th | 18.7 | 12.3 | | 19.5 | 12.4 | | 18.9 | 13.6 | | 22.3 | 15.4 | |
| 5th (least deprived) | 20.4 | 11.8 | | 19.8 | 10.4 | | 19.7 | 11.2 | | 21.8 | 12.7 | |
| **Region of residence** | | | p<0.001 | | | p<0.001 | | | p<0.001 | | | p<0.001 |
| North East | 4.9 | 4.9 | | 4.3 | 4.7 | | 4.2 | 4.9 | | 3.6 | 4.6 | |
| North West | 13.5 | 14.6 | | 12.0 | 13.6 | | 12.9 | 14.1 | | 11.2 | 13.5 | |
| Yorkshire and Humber | 10.3 | 11.3 | | 9.5 | 10.5 | | 9.5 | 10.1 | | 9.3 | 10.0 | |
| East Midlands | 9.1 | 8.1 | | 7.6 | 8.2 | | 8.4 | 7.6 | | 8.2 | 8.5 | |
| West Midlands | 11.0 | 10.2 | | 9.4 | 12.1 | | 9.3 | 11.4 | | 9.4 | 11.5 | |
| East of England | 10.9 | 10.3 | | 12.1 | 9.4 | | 11.5 | 10.4 | | 12.2 | 10.5 | |
| London | 12.5 | 19.0 | | 17.2 | 20.9 | | 17.1 | 19.7 | | 17.3 | 20.8 | |
| South East | 17.8 | 13.6 | | 17.8 | 13.0 | | 17.4 | 14.5 | | 19.3 | 14.2 | |
| South West | 10.0 | 7.8 | | 10.0 | 7.6 | | 9.7 | 7.5 | | 9.6 | 6.4 | |

*(Continued)*

**Table 1.** (Continued)

| Year of survey | 2006 | | | | 2010 | | | | 2014 | | | | 2018 | | |
| --- | --- | --- | --- | --- | --- | --- | --- | --- | --- | --- | --- | --- | --- | --- | --- |
| | Respondents (N = 2,966) | Non-respondents (N = 1,865) | p value | | Respondents (N = 5,332) | Non-respondents (N = 4,526) | p value | | Respondents (N = 4,568) | Non-respondents (N = 5,218) | p value | | Respondents (N = 4,509) | Non-respondents (N = 11,019) | p value |
| **Sex of baby** | | | p = 0.789 | | | | | | | | p = 0.622 | | | | p = 0.959 |
| Male | 52.2 | 51.8 | | | NA | NA | | | 50.9 | 51.4 | | | 51.3 | 51.3 | |
| Female | 47.8 | 48.2 | | | NA | NA | | | 49.1 | 48.6 | | | 48.7 | 48.7 | |
| **Multiplicity** | | | | | | | | | | | p = 0.876 | | | | p = 0.763 |
| Single birth | NA | NA | | | NA | NA | | | 98.5 | 98.4 | | | 97.6 | 97.5 | |
| Multiple birth | NA | NA | | | NA | NA | | | 1.5 | 1.6 | | | 2.4 | 2.5 | |
| **Parity** | | | | | | | | | | | | | | | p<0.001 |
| Primiparous | NA | NA | | | NA | NA | | | NA | NA | | | 51.5 | 38.6 | |
| Multiparous | NA | NA | | | NA | NA | | | NA | NA | | | 48.5 | 61.4 | |

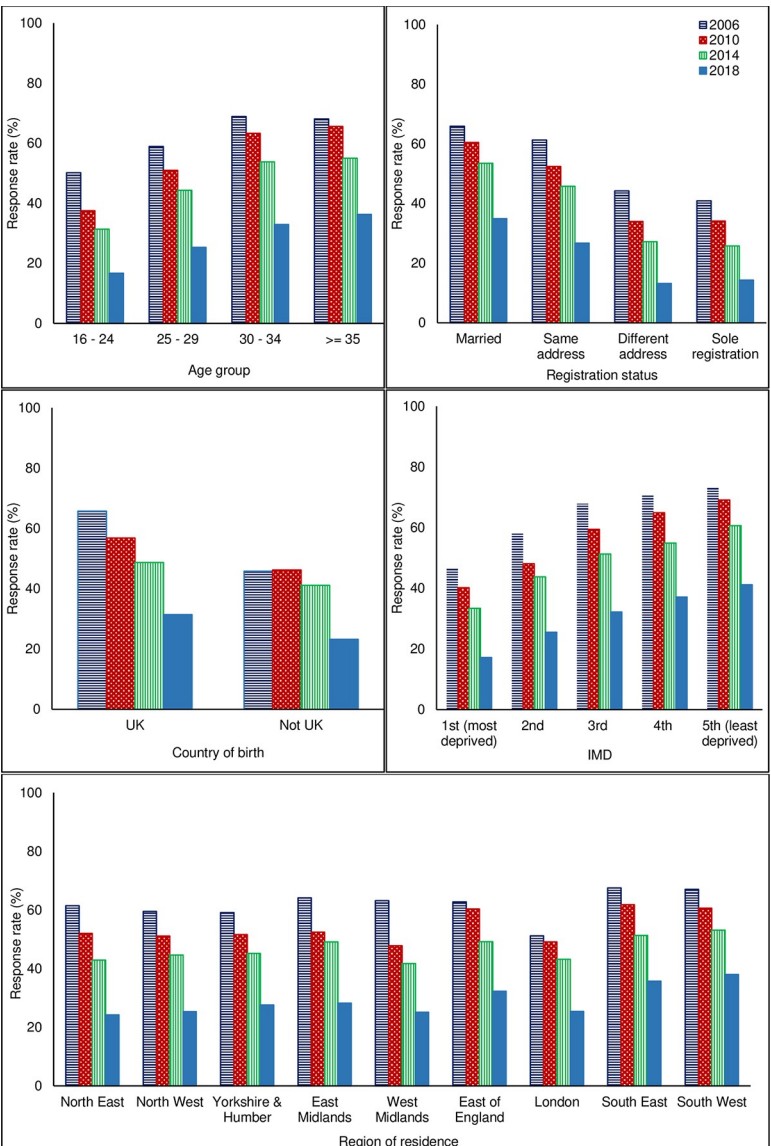

**Fig 1. Bar charts to show the association between demographic characteristics and response to the 2006, 2010, 2014 and 2018 national maternity surveys.**

estimates (<5.0%), although it is important to note that the breastfeeding initiation estimate is based on a subset of births in England, unlike the estimates for caesarean section, preterm birth, and low birthweight, which are based on all births in England. The proportions of missing data are also low for the estimates across all of the national maternity surveys (<4.3%).

The external validity of the survey estimates varied by maternity indicator and across surveys. For caesarean section, both the unweighted and weighted survey estimates were close to the population-based estimates and the 95% CIs included the population-based estimate for all surveys (Fig 2). For preterm birth, the unweighted survey estimates were lower than the population-based estimates and the 95% CIs did not include the population-based estimate for the 2006, 2010 or 2014 surveys. The effect of weighting on the 2014 and 2018 survey estimates was a marginal shift towards the population-based estimates and the 95% CIs for the weighted

**Table 2. Crude and adjusted odds ratios with 95% confidence intervals for association between demographic characteristics and response to the 2014 or 2018 surveys.**

| Year of survey (sample size) | 2014 (N = 9,786) | | | 2018 (N = 15,528) | | |
|---|---|---|---|---|---|---|
| | **OR** | **AOR** | **AOR 95%CI** | **OR** | **AOR** | **AOR 95%CI** |
| **Age group** (p-value) | (<0.001) | (<0.001) | | (p<0.001) | (p<0.001) | |
| <25 years | 0.39 | 0.56 | 0.49, 0.64 | 0.41 | 0.50 | 0.44, 0.58 |
| 25–29 years | 0.68 | 0.78 | 0.70, 0.86 | 0.69 | 0.74 | 0.68, 0.82 |
| 30–34 years | 1 | 1 | | 1 | 1 | |
| 35+ years | 1.05 | 1.04 | 0.92, 1.16 | 1.16 | 1.27 | 1.15, 1.39 |
| **Marital status**(p-value) | (<0.001) | (p<0.001) | | (p<0.001) | (p<0.001) | |
| Married | 1 | 1 | | 1 | 1 | |
| Joint names (same address) | 0.74 | 0.76 | 0.68, 0.84 | 0.68 | 0.68 | 0.62, 0.74 |
| Other | 0.32 | 0.40 | 0.34, 0.45 | 0.29 | 0.38 | 0.33, 0.44 |
| **Place of birth** (p-value) | (<0.001) | (p<0.001) | | (p<0.001) | (p<0.001) | |
| UK | 1 | 1 | | 1 | 1 | |
| Bangladesh | 0.33 | 0.30 | 0.20, 0.45 | 0.28 | 0.32 | 0.20, 0.51 |
| India | 0.72 | 0.54 | 0.41, 0.72 | 0.50 | 0.39 | 0.29, 0.53 |
| Pakistan | 0.42 | 0.38 | 0.29, 0.49 | 0.31 | 0.31 | 0.23, 0.43 |
| Africa | 0.60 | 0.57 | 0.47, 0.69 | 0.49 | 0.51 | 0.42, 0.62 |
| Europe | 1.03 | 0.93 | 0.80, 1.07 | 0.87 | 0.79 | 0.71, 0.89 |
| Other | 0.78 | 0.60 | 0.50, 0.73 | 0.73 | 0.58 | 0.49, 0.69 |
| **IMD** (p-value) | (<0.001) | (p<0.001) | | (p<0.001) | (p<0.001) | |
| 1st–most deprived | 0.33 | 0.50 | 0.43, 0.57 | 0.30 | 0.54 | 0.47, 0.61 |
| 2nd | 0.51 | 0.66 | 0.58, 0.76 | 0.50 | 0.72 | 0.64, 0.81 |
| 3rd | 0.69 | 0.82 | 0.71, 0.94 | 0.68 | 0.83 | 0.74, 0.93 |
| 4th | 0.79 | 0.87 | 0.75, 1.01 | 0.84 | 0.93 | 0.83, 1.05 |
| 5th–least deprived | 1 | 1 | | 1 | 1 | |
| **Region of residence** (p-value) | | (<0.001) (0.524)* | | (p<0.001) | (p<0.001) | |
| North East | 0.99 | 1.08 | - | 0.94 | 1.06 | 0.86, 1.30 |
| North West | 1.05 | 1.04 | - | 0.99 | 1.05 | 0.91, 1.21 |
| Yorkshire & Humber | 1.08 | 1.11 | - | 1.12 | 1.17 | 1.01, 1.36 |
| East Midlands | 1.27 | 1.14 | - | 1.15 | 1.12 | 0.96, 1.31 |
| West Midlands | 0.94 | 1.02 | - | 0.99 | 1.07 | 0.93, 1.24 |
| East of England | 1.28 | 1.01 | - | 1.40 | 1.20 | 1.04, 1.38 |
| London | 1 | 1 | - | 1 | 1 | |
| South East | 1.38 | 1.04 | - | 1.63 | 1.31 | 1.15, 1.48 |
| South West | 1.49 | 1.21 | - | 1.80 | 1.51 | 1.29, 1.77 |
| **Sex of baby** (p-value) | (0.622)^ | | | (0.959)^ | | |
| Male | 1 | - | - | 1 | - | - |
| Female | 1.02 | - | - | 1.01 | - | - |
| **Multiplicity** (p-value) | (0.876)^ | | | (0.763)^ | | |
| Single birth | 1 | - | - | 1 | - | - |
| Multiple birth | 0.98 | - | - | 0.97 | - | - |
| **Parity** (p-value) | | | | (p<0.001) | (p<0.001) | |
| Primiparous | NA | NA | | 1 | 1 | |
| Multiparous | NA | NA | | 0.59 | 0.52 | 0.49, 0.57 |

^ Variable not significantly associated with response in the univariable analysis.

* Variable not significantly associated with response in the multivariable analysis.

**Table 3. Unweighted and weighted distributions of respondent demographic characteristics used to calculate the weights in the 2014 and 2018 national maternity surveys (compared to overall sample).**

| Year of survey | 2014 | | | 2018 | | |
|---|---|---|---|---|---|---|
| | Respondents (N = 4,568) | Respondents (N = 4,568) | ONS Sample (N = 9,786) | Respondents (N = 4,509) | Respondents (N = 4,509) | ONS Sample N = 15,528 |
| | unweighted % | weighted % | % | unweighted % | weighted % | % |
| **Age** | | | | | | |
| <25 years | 14.0 | 21.0 | 20.8 | 9.3 | 16.3 | 16.1 |
| 25–29 years | 26.9 | 28.3 | 28.4 | 23.4 | 26.9 | 26.8 |
| 30–34 years | 34.7 | 30.1 | 30.1 | 38.0 | 33.4 | 33.5 |
| 35+ years | 24.4 | 20.6 | 20.7 | 29.3 | 23.3 | 23.5 |
| **Marital status** | | | | | | |
| Married | 60.1 | 52.0 | 52.5 | 63.5 | 52.8 | 52.8 |
| Joint names (same address) | 30.5 | 31.1 | 31.1 | 29.3 | 31.6 | 31.9 |
| Other | 9.4 | 16.9 | 16.5 | 7.1 | 15.6 | 15.3 |
| **Place of birth** | | | | | | |
| UK | 76.3 | 72.5 | 73.1 | 77.2 | 70.9 | 71.4 |
| Bangladesh | 0.7 | 1.3 | 1.3 | 0.5 | 1.3 | 1.2 |
| India | 2.0 | 2.4 | 2.3 | 1.3 | 2.0 | 2.0 |
| Pakistan | 2.0 | 3.3 | 3.3 | 1.1 | 2.9 | 2.6 |
| Africa | 3.9 | 5.3 | 5.0 | 2.9 | 4.8 | 4.7 |
| Europe | 10.2 | 9.5 | 9.6 | 12.0 | 12.2 | 12.3 |
| Other | 4.9 | 5.6 | 5.4 | 5.1 | 5.9 | 5.9 |
| **IMD** | | | | | | |
| 1$^{st}$–most deprived | 19.5 | 27.6 | 27.3 | 15.7 | 27.0 | 26.4 |
| 2$^{nd}$ | 21.4 | 22.8 | 22.8 | 19.3 | 22.2 | 21.9 |
| 3$^{rd}$ | 20.4 | 18.7 | 18.6 | 21.0 | 18.4 | 18.9 |
| 4$^{th}$ | 18.9 | 15.9 | 16.1 | 22.3 | 17.1 | 17.4 |
| 5$^{th}$–least deprived | 19.7 | 15.0 | 15.2 | 21.8 | 15.2 | 15.4 |
| **Region of residence** | | | | | | |
| North East | 4.2 | * | * | 3.6 | 4.0 | 4.3 |
| North West | 12.9 | * | * | 11.2 | 12.8 | 12.9 |
| Yorkshire & Humber | 9.5 | * | * | 9.3 | 9.8 | 9.8 |
| East Midlands | 8.4 | * | * | 8.2 | 8.7 | 8.4 |
| West Midlands | 9.3 | * | * | 9.4 | 11.0 | 10.9 |
| East of England | 11.5 | * | * | 12.2 | 11.1 | 11.0 |
| London | 17.1 | * | * | 17.3 | 20.1 | 19.8 |
| South East | 17.4 | * | * | 19.3 | 15.3 | 15.7 |
| South West | 9.7 | * | * | 9.6 | 7.3 | 7.3 |
| **Parity** | | | | | | |
| Primiparous | - | - | - | 51.5 | 42.2 | 42.3 |
| Multiparous | - | - | - | 48.5 | 57.8 | 57.7 |

* Characteristic not used in the calculation of the weights.

- Data unavailable.

survey estimates included the population-based estimates (Fig 3). For low birthweight, the unweighted survey estimates were mostly lower than the population-based estimates and the 95% CIs did not include the population-based estimates for the 2006 or 2010 surveys. The

**Table 4. External validity of estimates of selected maternity indicators from unweighted and weighted national maternity survey data from 2006, 2010, 2014 and 2018.**

| Year of survey | 2006 Respondents (N = 2,966) unweighted % | 2006 Routine data % | 2010 Respondents (N = 5,332) unweighted % | 2010 Routine data* % | 2014 Respondents (N = 4,568) unweighted % | 2014 Respondents (N = 4,568) weighted % | 2014 Routine data % | 2018 Respondents (N = 4,509) unweighted % | 2018 Respondents (N = 4,509) weighted % | 2018 Routine data* % |
|---|---|---|---|---|---|---|---|---|---|---|
| Caesarean section | 22.8 (21.3, 24.4) Missing = 0.7% | 24.2[1] Missing = 5.0% | 24.7 (23.5, 25.9) Missing = 1.8% | 24.8[1] Missing = 2.6% | 26.4 (25.2, 27.8) Missing = 1.8% | 25.9 (24.6, 27.2) | 26.5[1] Missing = 1.4% | 29.2 (27.9, 30.5) Missing = 0.2% | 27.2 (26.0, 28.8) | 28.0[1] Missing = 2.0% |
| Pre-term birth (<37 weeks) | 5.6 (4.8, 6.5) Missing = 1.8% | 7.5[2] Missing = 0.7% | 6.5 (5.8, 7.2) Missing = 1.3% | 7.3[2] Missing = 1.0% | 6.6 (5.9, 7.4) Missing = 2.3% | 6.9 (6.2, 7.7) | 7.6[2] Missing = 0.5% | 7.2 (6.5, 8.0) Missing = 1.4% | 7.6 (6.9, 8.4) | 8.0[2] Missing = 0.3% |
| Low birthweight (<2500 grams) | 5.8 (5.0, 6.7) Missing = 1.3% | 7.6[3] Missing = 1.0% | 5.4 (4.8, 6.1) Missing = 4.3% | 7.2[3] Missing = 0.7% | 7.5 (6.8, 8.4) Missing = 2.7% | 8.2 (7.4, 9.0) | 7.1[3] Missing = 1.6% | 6.8 (6.1, 7.6) Missing = 3.1% | 7.2 (6.4, 8.0) | 7.1[3] Missing = 1.9% |
| Breastfeeding initiation | 80.2 (78.8, 81.7) Missing = 1.0% | 66.2[4] (69.1 adjusted) Missing = 4.2% | 83.7 (82.6, 84.7) Missing = 2.4% | 71.7[4] (72.8 adjusted) Missing = 1.5% | 86.7 (85.7, 87.7) Missing = 2.8% | 84.5 (83.4, 85.6) | 74.0[4] (75.5 adjusted) Missing = 2.1% | 88.9 (88.0, 89.8) Missing = 0.3% | 85.3 (84.3, 86.4) | 74.5[4] (75.7 adjusted) Missing = 1.5% |

All published data are for live births.

[1] NHS Digital Hospital Episode Statistics (England).

[2] ONS Birth characteristics (England and Wales).

[3] ONS Birth characteristics (England).

[4] NHS Digital Maternity Services Dataset and Department of Health* Routine data for 2009 and 2017.

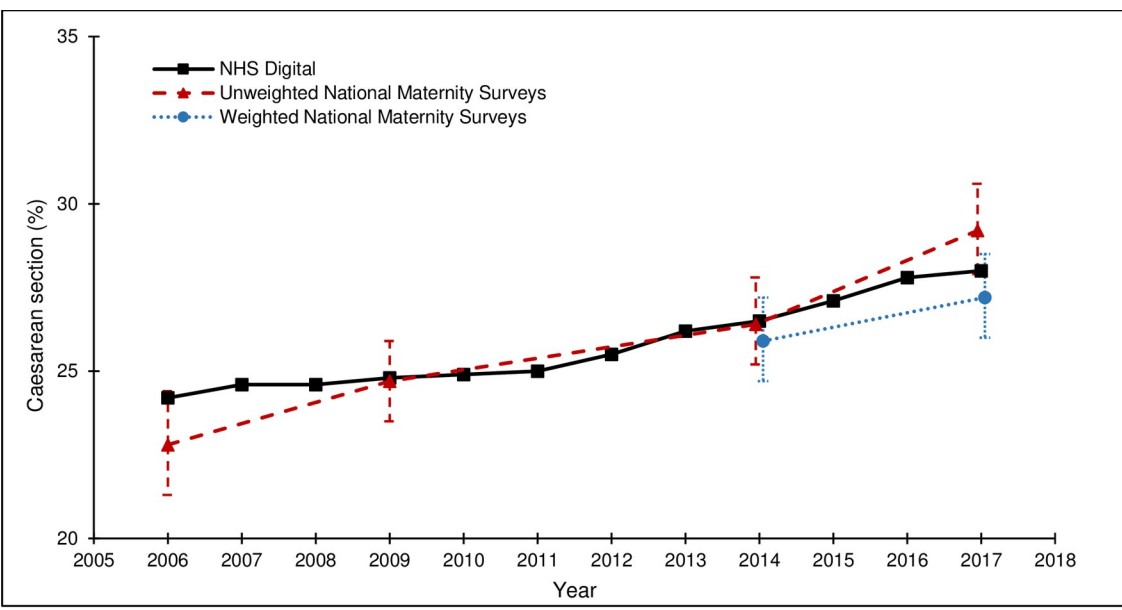

**Fig 2. Caesarean section rates from the national maternity surveys^ and NHS Digital.** ^ Women gave birth in 2009, survey conducted in 2010; Women gave birth in 2017, survey conducted in 2018.

effect of the weighting on the 2014 survey estimate was a shift away from the population-based estimate (and the 95% CIs did not include the population-based estimate); the effect of the weighting on the 2018 survey estimate was a shift towards the population-based estimate (and the 95% CIs included the population-based estimate) (Fig 4). For breastfeeding initiation, the unweighted survey estimates were higher than the population-based estimates and the 95% CIs did not include the population-based estimate for any survey (Fig 5). The effect of the

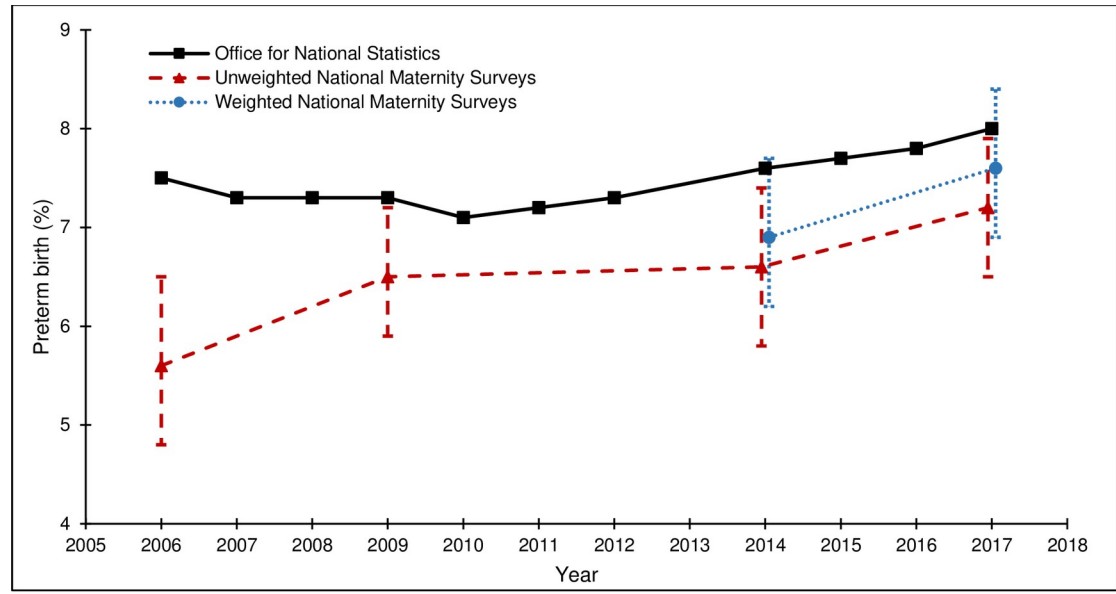

**Fig 3. Pre-term birth rates from the national maternity surveys^ and ONS.** ^ Women gave birth in 2009, survey conducted in 2010; Women gave birth in 2017, survey conducted in 2018.

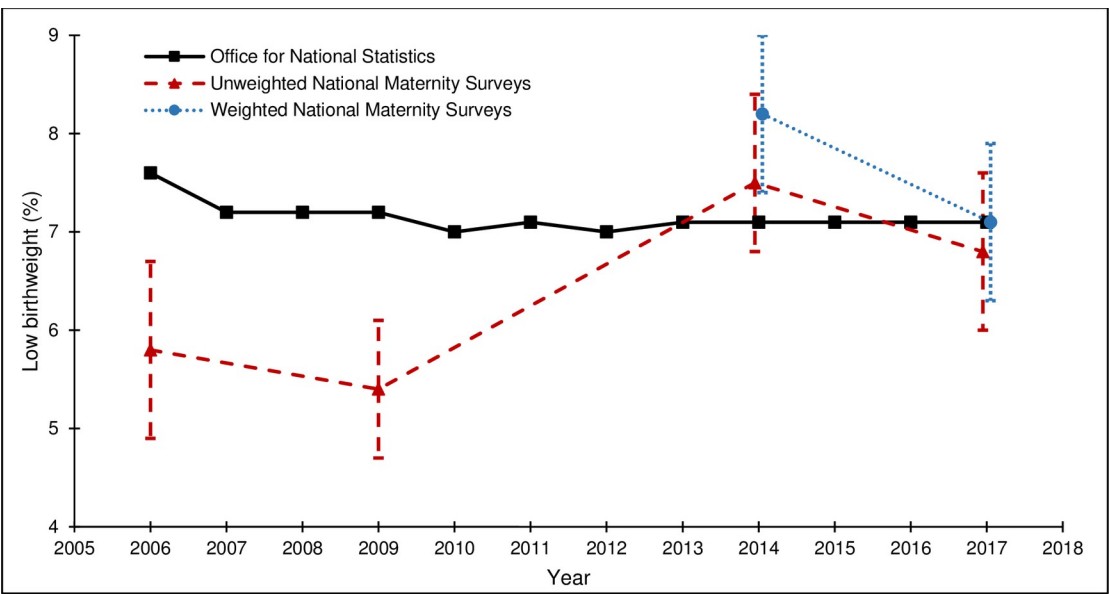

**Fig 4. Low birthweight rates from the national maternity surveys^ and ONS.** ^ Women gave birth in 2009, survey conducted in 2010; Women gave birth in 2017, survey conducted in 2018.

weighting on the 2014 and 2018 survey estimates was a shift towards the population-based estimates, yet the weighted survey estimates were still higher than the population-based estimates and the 95% CIs did not include the population-based estimates.

Despite the variability in response rates across the four surveys, there were no clear differences in the external validity of the unweighted or weighted survey estimates, compared to the population-based estimates from routine data.

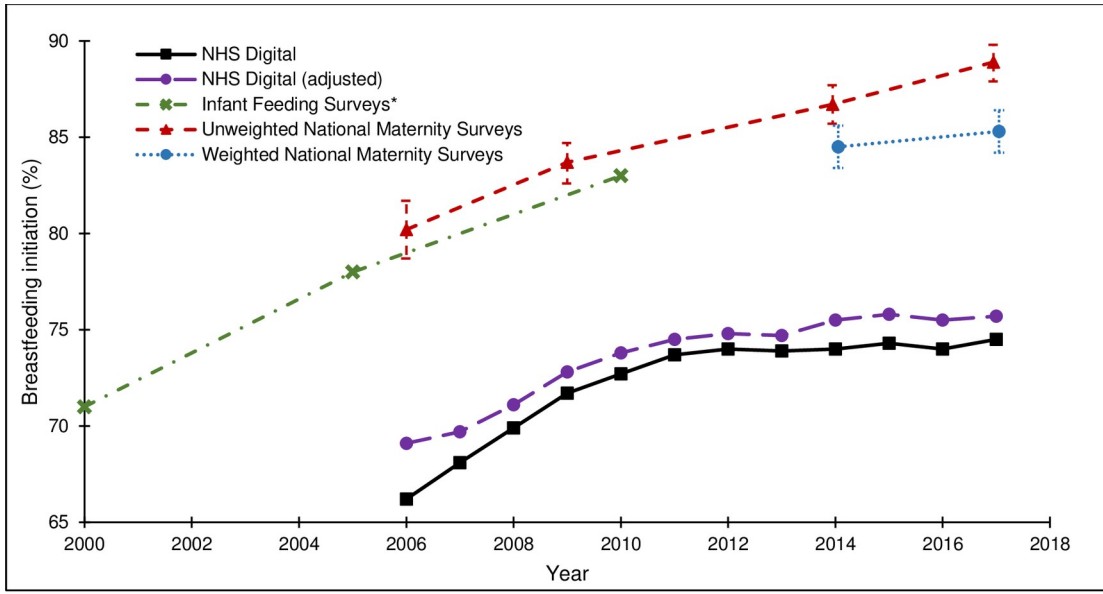

**Fig 5. Breastfeeding initiation rates from the national maternity surveys^, IFS and NHS Digital.** ^ Women gave birth in 2009, survey conducted in 2010; Women gave birth in 2017, survey conducted in 2018. * IFS data for 2005 and 2010 refers to England only; data for 2000 combines England and Wales.

## Discussion

The results of this secondary analysis of data from the national maternity surveys in England show that the respondents differed to the non-respondents on key demographic characteristics in each of the surveys carried out between 2006 and 2018. Respondents to all surveys were more likely to be older, married when they registered the birth of their baby, born in the UK, living in more advantaged areas, and living in the South or East of England. In 2018, first-time mothers were also more likely to respond. The survey response rate decreased across all of the demographic subgroups with each successive survey, with the most marked decrease in the most recent 2018 survey.

Despite the declining response rates, the unweighted survey estimates for some of the selected maternity indicators were relatively close to the population-based estimates from national routine data. The unweighted survey estimates of caesarean section rates from all of the surveys were close to the population-based estimates from NHS Digital. The unweighted survey estimates for preterm birth and low birthweight from most of the surveys were only slightly lower than the population-based estimates from ONS. Therefore, the findings suggest that prevalence estimates can still be valid even when survey response rates are low. Indeed, an extensive review of the literature found widespread evidence that changes in non-response rates rarely have large effects on survey estimates, with non-response rate being a lesser threat to survey estimates than is often indicated [12–15, 23]. However, the current results also found that the unweighted survey estimates of breastfeeding initiation from all of the surveys were considerably higher than the population-based estimates from NHS Digital (and formerly DH).

The results show that the survey weights mostly increased the validity of the 2014 and 2018 survey estimates of the maternity indicators, when evaluated against population-based estimates from national routine data. The effect the weighting had on the survey estimates differed according to the maternity indicator. For the indicators where the unweighted survey estimates were already close to the population-based estimates, the effect of the weighting was minimal. For the breastfeeding indicator, where the unweighted survey estimates were considerably higher than the population-based estimates, the effect of the weighting was a shift towards the population-based estimates, yet the weighted survey estimates were still higher than the population-based estimates. The effect of the weighting on the survey estimates was similar for both the 2014 (response rate 47%) and 2018 (response rate 29%) surveys. Therefore, the current findings do not provide evidence to suggest that survey weights perform better with higher response rates–or worse with lower response rates. To this end, the use of survey weights might be a method that could be useful if response rates to surveys continue to decline.

The current results are consistent with findings from other studies, which have examined the representativeness of survey respondents [10], the validity of prevalence estimates from survey data [12, 13], and the impact of survey weighting on the survey results [11]. The under- or over-representation of certain population groups within samples of survey respondents is a common problem in population-based surveys. The UK Biobank participants were found to be unrepresentative of the sampling population with evidence of a "healthy volunteer" selection bias [10]. Consequently, rates of all-cause mortality and total cancer incidence were significantly lower in subgroups of Biobank participants compared to equivalent subgroups in the general population. The National Survey of Sexual Attitudes and Lifestyles (NATSAL-3) in Britain found interview participants were somewhat higher-risk than the general population for all sexually transmitted infections (STIs), reflecting the oversampling of higher-risk individuals. In line with the current findings, the estimated prevalence of STIs shifted towards

estimates in the general population once interview sampling and non-response weights were applied, although for most STIs, the weights did not substantially affect prevalence estimates [11]. Some studies have reported greater success with weighting. The Belgian health interview survey found that, for most of the selected indicators, the use of post-stratification weights had a substantial impact on the prevalence of the estimates [24]. Finally, a UK study looking at efforts to reduce the risk of non-response bias found that appropriate weighting can remove much of the marginal non-response bias but the extent differs between variables. Broadly speaking, weighting corrected for non-response in the case of health and demographic variables, but not for attitudinal variables [25].

The population-based estimates of caesarean birth, preterm birth and low birthweight derived from routine data are considered to be reliable 'gold standard' estimates and the unweighted survey estimates of these maternity indicators were relatively close to the population-based estimates. Although the survey estimates of preterm birth and low birthweight were slightly lower than the population-based estimates, it should be noted that the ONS estimates are for all live births whereas the survey estimates are for babies who are alive at three to six months. Consequently, we would expect the survey estimates of preterm birth and low birthweight to be slightly lower than the ONS estimates because more of the babies who were born preterm and/or with low birthweight will have died by three to six months of age. The population-based estimates of breastfeeding initiation are based on datasets which have known issues with the completeness and quality of the data. Assuming that all infants whose breastfeeding status is unknown were not breastfeeding is particularly problematic and is likely to lead to underestimates of breastfeeding initiation rates. Therefore, although the unweighted survey estimates of this maternity indicator were not close to the population-based estimates, the caveats with the routine data mean that caution is required when interpreting the external validity of the breastfeeding survey estimates, based solely on comparison with the population-based estimates. It is noteworthy that the Infant Feeding Surveys (IFS), large UK surveys of infant feeding practice which provided data on prevalence of breastfeeding initiation until 2010, reported estimates more consistent with the estimates from the national maternity surveys [3].

The differences in the external validity of the survey estimates of selected maternity indicators may also suggest that non-response bias can affect some indicators more than others. Non-response bias might affect estimates of behavioural outcomes, such as breastfeeding, where women can exert relatively more control and choice (within medical and social constraints), to a greater extent than estimates of clinical outcomes such as mode of birth, gestational age and birthweight, which are largely uncontrollable. For example, the factors that influence whether or not a woman initiates breastfeeding may also influence whether or not a woman responds to a maternity survey. Another possible explanation is that questions on breastfeeding initiation may be more prone to reporting bias, due to social desirability, than questions on clinical outcomes such as caesarean section.

In our study, the weighting moved the 2014 and 2018 survey estimates towards the population-based estimates for the majority of the maternity indicators, yet the effect of the weighting was insufficient to bring all of the weighted survey estimates into line. This may be due to the information available on the sampling frame and the inclusion of limited demographic characteristics in the calculation of the weights. While some demographic characteristics associated with non-response could be measured and accounted for, other important factors may be unobserved and continue to bias prevalence estimates [11]. Therefore, weighting can reduce bias introduced by non-response, but it cannot eliminate bias because the reasons individuals participate in surveys are multifaceted. Older, married women who are living in more advantaged areas may be more likely to respond. Less observably, women who are more conscious

of health-related issues, have a greater sense of social responsibility, are comfortable disclosing personal information, or have more time at their disposal may also be more likely to respond. Such survey biases will not be eliminated by weighting for demographic characteristics [25].

The main strength of this study is that it compares prevalence estimates from four large national maternity surveys with similar methodology but different rates of response, allowing the effect of response rate on the validity of prevalence estimates from survey data to be examined. An additional strength is that the surveys all collected data on a number of key maternity indicators, enabling comparison with routinely collected data on these same indicators. A limitation of the study is that we were not able to consider additional maternity indicators, either because they were not collected in the survey questionnaires or because routine data are not available for comparison. A further limitation is that only aggregate demographic data were available for the women selected for the 2006 and 2010 surveys and so it was not possible to calculate survey weights for these survey data. Consequently, we were only able to look at the effect of the weights on prevalence estimates for the later surveys. Further work is needed to evaluate the external validity of prevalence estimates from survey data using different datasets and comparing the estimates with other reliable sources of external validation. Further work is also required to assess the effect of weighting on prevalence estimates from surveys, using different and/or additional variables in the calculation of the weights. The current findings indicate that survey weights do not provide a fully satisfactory solution to declining response rates and the effects of non-response bias and so efforts must continue to develop alternative methodological and statistical strategies to ensure that survey data are robust and prevalence estimates derived from survey data are valid and reliable.

## Conclusion

The external validity of the prevalence estimates from the survey data varied for the different maternity indicators and across the surveys but there were no clear differences according to survey response rate, suggesting that prevalence estimates may still be valid even when survey response rates are low. The survey estimates tended to become closer to the population-based estimates when survey weights were applied to the 2014 and 2018 survey data yet, for breastfeeding initiation, the effect was insufficient. However, it should be noted that the sources of the national routine data vary and the reliability of some published maternity indicators is uncertain. Survey weights may be useful when response rates are low, but they do not provide a fully satisfactory solution to declining response rates and the effect of non-response bias.

## Acknowledgments

Most thanks are due to the many women who participated in the national maternity surveys and to the women who provided input into the development of the questionnaire. Staff at the Office for National Statistics drew the samples and managed the mailings but bear no responsibility for analysis or interpretation of the data. Ciconi printed and prepared the survey packs and were responsible for the data entry. Qualtrics and Ciconi (in the 2018 survey) set up the online surveys.

## Author Contributions

**Conceptualization:** Sian Harrison, Maria A. Quigley.

**Data curation:** Sian Harrison.

**Formal analysis:** Sian Harrison, Maria A. Quigley.

**Funding acquisition:** Fiona Alderdice, Maria A. Quigley.

**Investigation:** Sian Harrison, Fiona Alderdice, Maria A. Quigley.

**Methodology:** Sian Harrison, Fiona Alderdice, Maria A. Quigley.

**Project administration:** Sian Harrison.

**Writing – original draft:** Sian Harrison.

**Writing – review & editing:** Sian Harrison, Fiona Alderdice, Maria A. Quigley.

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
