## [Decision Letter · Decision Letter 0]

25 Aug 2020

PONE-D-20-18902

External validity of prevalence estimates from the National Maternity Surveys in England: the impact of response rate

PLOS ONE

Dear Dr. Harrison,

Thank you for submitting your manuscript to PLOS ONE. After careful consideration, we feel that it has merit but does not fully meet PLOS ONE’s publication criteria as it currently stands. Therefore, we invite you to submit a revised version of the manuscript that addresses the points raised during the review process.

We look forward to receiving your revised manuscript.

Kind regards,

Diane Farrar

Academic Editor

PLOS ONE

2. In your Methods section, please provide a link for every data source used.

**Comments to the Author**

1. Is the manuscript technically sound, and do the data support the conclusions?

Reviewer #1: Yes

Reviewer #2: Yes

2. Has the statistical analysis been performed appropriately and rigorously? 

Reviewer #1: Yes

Reviewer #2: Yes

3. Have the authors made all data underlying the findings in their manuscript fully available?

Reviewer #1: Yes

Reviewer #2: Yes

4. Is the manuscript presented in an intelligible fashion and written in standard English?

Reviewer #1: Yes

Reviewer #2: Yes

5. Review Comments to the Author

Reviewer #1: This is an extremely important paper. As survey response rates have been dropping, the field of survey research and users of survey data are very interested in whether lower response rates indicate lower accuracy of survey measurements. This paper uses unique and impressive data to do the type of analysis that is extremely difficult for most surveys: to benchmark against official records that are likely to be highly accurate. The findings, that lower respondents rates do not indicate lower accuracy, align with a few other papers exploring the same sorts of issues with very different types of data. Therefore, this paper is extremely valuable at this time and will be very highly cited if it's made sufficiently visible. Also, the paper is really helpful by exploring whether conventional weighting increases accuracy of survey estimates - the evidence of improved accuracy due to weighting is important, because this methodology has usually been taken for granted and has rarely been subjected to empirical testing. So this testing is very helpful. Also helpful is the evidence that the accuracy of these surveys was generally extremely high - that, too, will cause the paper to be heavily cited, for good reason. And the evidence here that "true" rates often fell outside the margin of error of the survey estimate is a useful wake-up call telling researchers that these margins of error are most likely misleadingly small.

I commend the authors for working on an extremely important topic, using impressive data, doing thorough and thoughtful analyses, and generating extremely valuable results.

I have two suggestions: First, the authors should cite more thoroughly the small number of previous publications exploring the impact of response rates on survey accuracy, including Curtin et al. (POQ), Keeter et al. (POQ), Merkle and Edelman (book chapter), Holbrook et al. (book chapter).

Second, and more serious, I appreciate the authors' discussion of breastfeeding, but there's a problem here. Treating unknown breastfeeding status as not breastfeeding no doubt (as the authors say) biases those numbers downward from the routine data. And unlike the other measures, I presume that even the routine data for breastfeeding are based on parent reports rather than more objective data. So I would strongly suggest a different approach to breastfeeding: (1) be much more explicit with readers about how the routine data measurements were made, (2) tell readers the proportions of babies for whom breastfeeding was not known in each survey, (3) if those numbers are large enough, consider dropping the breastfeeding data from the paper altogether, (4) if those data are not dropped, discuss them completely separately from all other data, making it much clearer to readers that the routine data on breastfeeding may be problematic enough to undermine the value of the analyses with that variable.

Reviewer #2: This paper investigates the accuracy of population estimates derived from the National Maternity Survey across varying levels of response rates obtained between 2006 and 2018. Survey estimates are compared with gold standard with information obtained NHS Digital. For the two most recent surveys, weighted and unweighted survey data can both be compared. The analyses conclude that there is no association between accuracy of estimates and survey response rates. The paper is well-done. The writing is clear, the data are of high quality, the statistical analyses are excellent, and the findings make good sense. A valuable finding and conclusion is that the accuracy of survey estimates seem to vary by question topic. I frankly see little to quibble with in this manuscript and believe it will be a useful contribution, specifically to the field of health survey research methodology.

6. PLOS authors have the option to publish the peer review history of their article (what does this mean?). If published, this will include your full peer review and any attached files.

Reviewer #1: No

Reviewer #2: No

---

## [Author Response · Author response to Decision Letter 0]

5 Nov 2020

Please find below our response to each point raised by the editor and reviewers. 

Academic Editor

The manuscript has been revised in line with PLOS ONE’s style requirements using the templates provided. The references and the file names of the manuscript and the figures have also been updated. 

In your Methods section, please provide a link for every data source used.

References to each of the reports for the previous national maternity surveys in 2006, 2010, 2014 and 2018 have been added to the methods section and the references section has been updated accordingly. 

Reviewer 1

First, the authors should cite more thoroughly the small number of previous publications exploring the impact of response rates on survey accuracy, including Curtin et al. (POQ), Keeter et al. (POQ), Merkle and Edelman (book chapter), Holbrook et al. (book chapter).

We thank the reviewer for the positive review and for the suggestions for additional references. The manuscript has been revised and all of the above publications are now cited.

Second, and more serious, I appreciate the authors' discussion of breastfeeding, but there's a problem here. Treating unknown breastfeeding status as not breastfeeding no doubt (as the authors say) biases those numbers downward from the routine data. And unlike the other measures, I presume that even the routine data for breastfeeding are based on parent reports rather than more objective data. So I would strongly suggest a different approach to breastfeeding: (1) be much more explicit with readers about how the routine data measurements were made, (2) tell readers the proportions of babies for whom breastfeeding was not known in each survey, (3) if those numbers are large enough, consider dropping the breastfeeding data from the paper altogether, (4) if those data are not dropped, discuss them completely separately from all other data, making it much clearer to readers that the routine data on breastfeeding may be problematic enough to undermine the value of the analyses with that variable.

Thank you to the reviewer for raising this important point and for the helpful suggestions about how to approach the issue. We have made the following revisions:

• We have stated clearly how the prevalence of breastfeeding initiation status is measured in the routine data (page 8, lines 202-212 on revised manuscript with track changes).

• We have examined the proportion of women with missing data for each of the maternity indicators assessed in the surveys. 

• The proportion of women with unknown breastfeeding initiation status was low across surveys (<4.3%) and so we have decided to include the survey data on this indicator in the analysis. However, the proportion of women with unknown breastfeeding status at 6-8 weeks was high, at least in the 2014 survey (25.0%). Therefore, in line with the reviewer’s recommendation, we have decided to exclude the breastfeeding status at 6-8 weeks indicator from the analysis. 

• We have stated the proportion of missing data for each of the indicators across all of the surveys and for the routine data (page 15, table 4 on revised manuscript with track changes). This is 5% or less for all indicators.

• We have revised the discussion to make clearer the limitations with the routine data on breastfeeding initiation status and the potential impact of this on the validity of the analysis of the breastfeeding data (page 19, lines 430-432 on revised manuscript with track changes).

Reviewer 2

We thank the reviewer for the positive review and note there are no points to which we need to respond.

---

## [Editor Report · Decision Letter 1]

10 Nov 2020

External validity of prevalence estimates from the National Maternity Surveys in England: the impact of response rate

PONE-D-20-18902R1

Dear Dr. Harrison,

We’re pleased to inform you that your manuscript has been judged scientifically suitable for publication and will be formally accepted for publication once it meets all outstanding technical requirements.

Kind regards,

Diane Farrar

Academic Editor

PLOS ONE

---

## [Editor Report · Acceptance letter]

13 Nov 2020

PONE-D-20-18902R1 

External validity of prevalence estimates from the National Maternity Surveys in England: the impact of response rate 

Dear Dr. Harrison:

I'm pleased to inform you that your manuscript has been deemed suitable for publication in PLOS ONE. Congratulations! Your manuscript is now with our production department. 

Kind regards, 

on behalf of

Dr. Diane Farrar 

Academic Editor

PLOS ONE